# Morphological Characterization of the Antenna and Scent Patch of Three *Danaus* Species (Papilionoidea: Nymphalidae, Danainae)

**DOI:** 10.3390/insects15020121

**Published:** 2024-02-07

**Authors:** Yaqi Yang, Linyun Ding, Tong Wang, Huaijian Liao, Chufei Tang

**Affiliations:** 1College of Biotechnology, Jiangsu University of Science and Technology, Zhenjiang 212100, China; yangyaq0123@163.com; 2Institute of Leisure Agriculture, Jiangsu Academy of Agricultural Sciences, Nanjing 210014, China; 3Central Laboratory, Jiangsu Academy of Agricultural Sciences, Nanjing 210014, China; 20210991@jaas.ac.cn (L.D.); wtcu@163.com (T.W.)

**Keywords:** *Danaus*, antenna, scent patch, androconial scale, sensilla, morphological characteristics

## Abstract

**Simple Summary:**

The scent system of *Danaus* is not only involved in sexual communication but also in mimicry. This study reports, for the first time, the morphological characteristics of *Danaus*’ antennae and scent patches of the scent system for three species and explores their correlations. The ultrastructure of different antennal segments and their sensillum types was investigated, as well as the hierarchical structure of the scent patch, particularly the ultrastructure of its scales. Significant correlations were reported between the morphological characteristics of androconial scales in scent patches. Moreover, the antennal characteristics were significantly correlated. The morphological characteristics of the females’ antennae were significantly correlated with those of the males’ antennae and androconial scales. However, the significance and coefficient of these correlations were inconsistent across species and sexes. The study provides fundamental morphological information that may be involved in the pheromone recognition system of *Danaus*.

**Abstract:**

The scent system of *Danaus* is important for the study of butterfly sexual communication and relevant investigations in biomimetics due to its involvement with mimicry. Using light, scanning, and transmission electron microscopy, the morphological characteristics of *Danaus*’ antennae and scent patches of the scent system for three species, *D. chrysippus*, *D. genutia*, and *D. plexippus*, were investigated herein. Their apical clubs of the flagellums contain sensilla trichodea, sensilla chaetica, and sensilla coeloconica. The scent patch scales typically have a tree-like structure in its lumen at the nano-scale. Comparisons were made between the androconial scales and the other scales in scent patches. Rank sum tests showed significant differences in scent patch scales’ characteristics between the species, as well as in the ultrastructure of antennal segments between species and sexes. Spearman’s correlation tests showed significant correlations between the morphological characteristics of androconial scales in scent patches. Moreover, the antennal characteristics were significantly correlated. The morphological characteristics of the females’ antennae were significantly correlated with those of the males’ antennae and androconial scales. However, the significance and coefficient of these correlations were inconsistent across species and sexes. This study provides fundamental morphological information that helps in understanding the pheromone recognition system of *Danaus*.

## 1. Introduction

*Danaus* Kluk (1780) is a cosmopolitan genus belonging to Danainae of Nymphalidae. It is one of the most abundant and remarkable butterfly genera, with 11 species, including monarchs, queens, tigers, and milkweed butterflies [1]. *Danaus* species exhibit typical Mullerian mimicry [2]. They are medium-to-large in size and are characterized by a bright and colorful wing pattern. The wings have a ground color of orange, yellow, or brown with bold black and white markings, which serve as a warning that the butterflies contain toxins in their bodies [3]. Like many Lepidopteran species, *Danaus* communicates with pheromones through the contact between scent patches and the apical club of antennae before mating, which is essential for successful courtship [4,5,6]. The structure of scent patches and antennae enables their function and therefore plays a crucial role in the recognition system [6,7]. So, the ultrastructure, abundance, and distribution of antennal sensilla, as well as the hierarchical characteristics of the scent patch like its thickness, are important in the pheromone recognition system of *Danaus* species [8,9]. For example, due to their periodic and multiporous ultrastructure, the androconial scales in the scent patches of *Danaus* males store and emit pheromones produced by glandular cells connected to them. These pheromones are detected and recognized via the sensilla on the females’ antennae. Specially, in *Danaus*, hairpencils located within the tip of the abdomen exclusively produce some critical pheromones for courtship [10]. These pheromones are transferred when the hairpencils dip into androconial pockets and are then stored and released like the other pheromones [10,11]. The complex scent organs of *Danaus* are suggested to be involved with their mimicry, as the visual similarity of their wing pattern may be a selective pressure for chemical communication, making them of great interest for studying the mechanisms and evolution of butterfly sexual communication [8]. Therefore, understanding the functional morphology of *Danaus*’ antennae and scent organs is crucial for studying butterflies.

The morphological characteristics of butterfly antennae and scent organs are useful in bionics, in addition to their biological significance. For example, androconial scales in scent patches also enable high cooling efficiency [12]. The androconial scales typically form a chitin layer with a relatively higher height. They improve IR absorption (emission) through the increased inner surface area and multi-scattering effect. This enhancement of IR emissivity enables the butterfly to efficiently radiate heat as well as the pheromone from the scent patch region to the environment [13]. The morphological characterization of antennae has been extensively used in the development of radar and robotic tactile technology [14,15]. Detailed descriptions of the taxonomic morphological characteristics of antennae and scent organs in *Danaus* species have received little attention [16]. Previous studies have provided ultrastructural characterizations of the antennae for *D. gilippus* (Cramer) and the scent organs of *D. chrysippus* (Linnaeus), which indicated homology in the ultrastructure of the scent patch and hairpencils [8,9]. However, no quantitative comparison was made between the morphological characteristics of the antennae and scent patches of the different *Danaus* species. It is also important to consider how the morphological characteristics of the antennae and scent patches fit together, as larger sensory organs can enhance sensitivity by increasing the number or size of sensing structures, but this often comes at a higher developmental cost [17]. However, there has been no investigation into the correlation between the morphological characterization of these two organs.

With this concern, the present study investigated the morphological characteristics of the antennae and scent patches for three *Danaus* species (*D. chrysippus* Linnaeus, *D. genutia* Cramer, and *D. plexippus* Linnaeus) using light microscopy, scanning electron microscopy (SEM), and transmission electron microscopy (TEM). The three species are similar in body size, but their scent patches are visually different in size, which benefits studying the correlations between the morphological characteristics. The differences in these characteristics between species and sexes were compared quantitatively using rank sum tests. Spearman’s correlation tests were utilized to examine the correlations between the morphological characteristics of the antennae and the androconial scales in scent patches.

## 2. Materials and Methods

### 2.1. Materials

*Danaus* specimens were obtained from natural populations using sweep nets. *D. chrysippus* and *D. genutia* were collected in July 2021 from Yuanjiang (Yunnan Province, China). *D. plexippus* was collected in May 2020 from Tingo Maria (Department of Huánuco, Peru). The species were identified on the basis of wing coloration and pattern [16]. All specimens were deposited at the Institute of Leisure Agriculture, Jiangsu Academy of Agricultural Sciences, China.

### 2.2. Microscopy

The morphological characteristics of the antenna were examined for both sexes. Both sides (dorsal and ventral) of the scent patches were investigated in males. Specimens were observed, photographed, and dissected under a stereoscopic microscope (Zeiss SteREO Discovery V.8, Gottingen, Germany). The surfaces of antennae and scent patches were investigated using SEM. Moreover, the sections of scent patches were examined using TEM. Six specimens were used for each sex of each species.

For SEM, the samples were washed with 0.1 mol/L Na-cacodylate buffer (pH 7.2) and then fixed in a 2.5% glutaraldehyde solution for 24 h. After fixation, the samples were dehydrated in a graded series of ethanol solutions (75%, 80%, 90%, and 95%) for 20 min at each gradient. The samples were soaked for 20 min at each gradient, and then preserved in 100% ethanol before drying with a critical point dryer (Leica EM CPD300, Wetzlar, Germany). Prior to observation, all dried samples were sputtered with gold. The samples were analyzed using a scanning electron microscope (Zeiss EVO-LS10, Gottingen, Germany) with an acceleration voltage range of 5–15 kV.

For TEM, the samples were fixed in 2.5% glutaraldehyde for 48 h, washed with 0.1 mol/L Na-cacodylate buffer (pH 7.2), postfixed in 1% osmic acid, and then dehydrated through a graded ethanol series (75%, 80%, 90%, and 95%), a 1:1 mixture of 95% ethanol solution and 95% acetone solution, 95% acetone solution, and 100% acetone. Each gradient was applied for 20 min, except for 100% acetone, where the samples were soaked for 40 min. The dehydrated samples were immersed in propylene and embedded in ethoxylate resin. Afterward, they were sectioned using an ultramicrotome (RMC Powertome XL, Tucson, AZ, USA) and observed with a transmission electron microscope (JEOL JEM-1400 Flash, Tokyo, Japan) at 120 kV.

### 2.3. Terminology

Sensilla were identified and categorized according to the terminology described by Boddum et al., Crook and Mordue, Schneider, and Limberger et al. [18,19,20,21]. The androconial scales in scent patches were described according to the wing scale terminology of Ghiradella [22].

### 2.4. Morphometric Measurements

All measurements were obtained from the photographs using FIJI-Image J (64-bit Java 1.8.0_172 [23]. The program can count the pixels occupied by a region or line and convert them to actual size using plotting scales. The polygon selection tool and the line tool were used to circumscribe the regions and lines to be measured, respectively. The measurements were applied as the mean of at least six individuals in each species of each sex.

In addition to the area of the scent patch and androconial scales, the following ultrastructural parameters that influence the efficiency of pheromone emission of the scent patch were measured quantitatively: (1) parameters positively correlated with emissivity, including the width of the crossrib, ridge and trabeculae, as well as the height of the ridge, crossrib, lower lamina and trabeculae; (2) parameters negatively correlated with emissivity, including the distance between the ridges, trabeculae and crossribs [24]. The areas of the scent patches were measured from microscopic photographs. The widths and distances between ridges and crossribs, in addition to the areas of scales, were measured from SEM photographs. The other parameters were measured from TEM photographs.

The following antennal parameters were measured: the size of the whole antenna and each antennal segment. Also, the number, length, and diameter of each sensillum type distributed on the club were measured, where the antenna touches the scent patches during pheromone recognition.

### 2.5. Data Analysis

Rank sum tests were conducted using IBM SPSS Statistics 2 to identify whether the quantitative parameters differed significantly between species (Kruskal–Wallis test) and sexes (Mann–Whitney U test). Spearman’s correlation tests were used to identify correlations between the morphological parameters of the antennae and the androconial scales in scent patches, as well as between the characteristics of male and female antennae. All statistical analyses were performed using R version 4.2.0. The R packages Hmisc, corrplot, and ggplot2 were used to perform the analyses and to visualize the results.

## 3. Results

### 3.1. Descriptions of the Wing Coloration and Pattern of the Three Danaus Species

The wings of the three *Danaus* species are predominantly orange in color (Figure 1). The front edge, tip, and outer edge of their forewings, as well as the outer edge of their hind wings, are dark-brown. *D. chrysippus* has four white spots on the sub-apex of the forewing, accompanied by several smaller white spots nearby. Additionally, there is a row of continuous small white spots on the apex and outer edge. On the hind wing, there is a dark-brown band on the outer margin, with one row of white spots and three dark-brown spots on the end of the discal cell. The wings of *D. genutia* are dark-brown from the front edge to the apex of the forewing. A white oblique band is present in the center, along with 1–2 columns of white spots. The wings of *D. plexippus* have two orange spots in the dark-brown area of the forewing apex and two rows of white spots in the outer margin of both the fore- and hind wings.

The three *Danaus* species exhibit sexual dimorphism in their general appearance. Although the wing patterns of the three species are highly similar between sexes, the males’ hind wings have a black scent patch located in the basal third of the Cu2 wing vein (Figure 1A–C), which is absent in the females (Figure 1D–F). The males are overall slightly smaller than the females for all species. The wingspan ranges from 6.8 to 7.6 cm and 7.0 to 7.8 cm for *D. chrysippus*, 7.6 to 8.0 cm and 7.6 to 8.2 cm for *D. genutia*, and 9.0 to 9.4 cm and 9.2 to 9.8 cm for *D. plexippus* in males and females, respectively.

### 3.2. Comparative Morphology of Antennae

Both sexes of the three species have clavate antennae that are black and long, with segments gradually widening towards the tip (Figure 1 and Figure 2). The three species have antennal lengths ranging between 25.66 ± 0.18 and 31.11 ± 0.18 mm. *D. plexippus* has a significantly greater antennal length compared to the other two species (Table 1). The scape length is between 0.35 ± 0.01 and 0.45 ± 0.01 mm and shows no significant differences between the target species. However, the range of pedicel length is between 0.29 ± 0.01 and 0.41 ± 0.01 mm and is significantly shorter in *D. genutia* than in the other two species. The length of the flagellum ranges from 24.98 ± 0.18 to 30.38 ± 0.18 mm, which is significantly larger in *D. plexippus* compared to the other two species. The flagellum of each species consists of 36–41 flagellomeres, with the apical club formed by 9–11 terminal flagellomeres (Figure 2). The number of flagellomeres varies among the three species: 36–37 in *D. genutia*, 37–41 in *D. chrysippus*, and 40–41 in *D. plexippus*. The length of the flagellomere is between 0.65 ± 0.02 and 0.75 ± 0.01 mm (Table 1). In comparison to the other two species, it is significantly greater in *D. plexippus*. The width of the apical club ranges from 0.73 ± 0.03 to 1.02 ± 0.02 mm, with the largest being in *D. plexippus* and the smallest in *D. genutia*, showing significant differences. The area of the apical club ranges from 3.18 ± 0.13 to 3.85 ± 0.28 mm^2^, with *D. genutia* having a significantly smaller area than the other two species. There are significant sexual differences in the lengths of the antenna, particularly the scape, pedicel, and flagellomere, as well as the area of the apical club. As shown in Table 1, most of these parameters are significantly larger in males than in females for all species. However, the width of the apical club does not differ significantly between sexes.

In both sexes of the three *Danaus* species, the flagellum contains three longitudinal ridges that gradually disappear from the scape towards the pedicel, forming two light grooves where most antennal sensilla and microtrichia are located (Figure 3A). The apical club contains three types of sensilla: sensilla trichodea (ST), sensilla chaetica (SCh), and sensilla coeloconica (SCo), in addition to the two subtypes of microtrichia (Figure 3B). Each flagellomere contains two oval depressions, the sulci, in which the ST and microtrichia subtype 1 (m1) are clustered. The sulci gradually become smaller and rounder towards the tip of the flagellum. The remaining regions of the grooves are densely covered by the microtrichia subtype 2 (m2), where the other two types of sensilla (SCh and Sco) are scattered. The two subtypes of microtrichia are leafy with distinct grooves and ridges (Figure 3C,D). The m1 is erect, short, and shrunken. The m2 is significantly longer, wider, and flatter than the m1 and pointed towards the antenna tip.

ST is the most abundant sensillum on the apical club (Figure 3D). It is long and hair-like with a smooth surface in the three *Danaus* species, with a length ranging from 17.40 ± 0.36 to 21.78 ± 0.49 μm and a basal diameter ranging from 1.77 ± 0.08 to 1.96 ± 0.06 μm (Table 1). The base of the sensillum is embedded in the fossa, with a curved tip always pointing towards the apex of the antenna. The average number of STs on each flagellomere ranges from 114 ± 1 to 131 ± 6 in the target species (Table 1). The number of STs in a sulcus varies in a single flagellomere.

The apical club of *Danaus* has SCh as the longest sensilla type, with a length ranging from 25.33 ± 1.92 to 34.20 ± 1.21 μm (Figure 3E, Table 1). These sensilla are straight or slightly curved bristles inclined at an angle of about 60° to the antenna’s surface. They are inserted in a socket with an articulated base surrounded by m1. The surface of SCh is rough-walled with longitudinal grooves that become flatter toward the blunt tip.

The SCo is bud-like, with an erect peg at the center of the hollow central concave and eight to ten petaloid-like pegs at the periphery, all inclined toward the central erect peg (Figure 3F). The central peg has a smooth surface, while the surrounding pegs are covered with crested longitudinal lines. All peg tips of SCo are directed toward the apex of the antenna. The length of SCo ranges from 15.49 ± 0.30 to 18.34 ± 0.28 μm, and the aperture diameter ranges from 8.27 ± 0.43 to 9.27 ± 0.19 μm (Table 1). Sco is distributed not only on the lateral side but also on the dorsum and venter of flagellomeres, with a higher density on the dorsum and lateral side than on the venter.

The number and size of sensilla show significant differences between species and sexes (Table 1). The number of STs per flagellomere is highest in *D. genutia* and lowest in *D. plexippus*. It is significantly higher in the males than in the females of *D. chrysippus* but higher in the females than in the males of *D. plexippus*; however, it does not show significant sexual differences in *D. genutia*. The length of the ST is significantly longer in *D. genutia* than in the other two species and always longer in the females than in the males in the three species. The diameter of the ST is significantly larger in *D. plexippus* than in *D. chrysippus*. It only has sexual differences in *D. plexippus*, in which it is larger in the females than in the males. The number of Sco is significantly higher in *D. genutia* compared with the other two species. It is also higher in the males than in the females in *D. plexippus* but does not differ significantly between the sexes in either *D. chrysippus* or *D. genutia*. With the longest length and the largest diameter, Sco is the largest in *D. plexippus.* The length of Sco is significantly longer in the females than the males in *D. genutia* and *D. plexippus*. The diameter of Sco is always larger in the females than the males in the three species. The number of SCh does not show a significant difference between the species or the sexes, but it is significantly longer and wider in *D. genutia* compared with the other two species. The length of SCh is significantly longer in the males than the females in *D. genutia.* The diameter of SCh is significantly larger in the females than the males in *D. plexippus*.

### 3.3. Comparative Morphology of Male Scent Patches

The scent patch, which is only present in males, is largest in *D. chrysippus* (19.85 ± 0.09 mm^2^), followed by *D. genutia* (4.82 ± 0.06 mm^2^), and smallest in *D. plexippus* (2.18 ± 0.03 mm^2^) (Figure 4). The rank sum tests show that there was a significant difference in scent patch size between any two of the three species (*p* < 0.001). The scent patch is similar on the dorsal wing surface in all three species, being foliated, overall humped, cupped in the middle, and wholly black (Figure 4A–F). But it differs in shape on the ventral surface of wings among the target species (Figure 4G–L). Specifically, in *D. plexippus*, the alar organ is smooth and lacks a distinct border, while in *D. chrysippus*, it is slightly thickened, rounded, and has a clear border. In *D. genutia*, the alar organ is foliated, with a clear border and a hump in the middle, corresponding to the region cupped on the dorsal wing surface. The color of scent patches on the ventral surface of wings also varies among the three species. It is wholly or mainly black in *D. plexippus* and *D. genutia*, while in *D. chrysippus*, it is white in the apical half. The scales in scent patches are arranged on the wing membrane like tiles on a roof at the micron scale.

On the dorsal surface of the wings of the three *Danaus* species, the scent patch scales are typically irregularly overlapped into 3–4 layers in the humped region and 2–3 layers in the surrounding area (Figure 5A–C). These scales are laminar but vary in shape and size. They are coronal with apexes that are mostly rounded, but occasionally bifurcated or trifurcated for all species. The scales in the humped regions are the androconial scales, with varied sizes; the largest is found in *D. genutia* (6315.24 ± 183.02 μm^2^), followed by *D. chrysippus* (5398.72 ± 210.48 μm^2^), with the smallest in *D. plexippus* (4815.82 ± 183.73 μm^2^). On the other hand, the scale density is highest in *D. plexippus*, followed by *D. chrysippus*, with the lowest in *D. genutia.* The density and size of scales are significantly different between the three species (*p* < 0.0001). The scales in the surrounding regions are similar in size to the androconial scales in each species, except for those in the groove between the humped region and the surrounding flat regions, which are about one-third the size of the androconial scales.

The scent patch scales on the ventral surface of the wings are arranged in regular rows and overlap into about two layers in the target species, which is typical for butterflies (Figure 5J–L). The ground scales are all coronal in the three *Danaus* species, but the shape of the scale apex is different between species. While the apex is rounded in *D. chrysippus*, it is trifurcated or quadrifurcated in *D. genutia*, and mostly trifurcated in *D. plexippus*. The size of these scales is 11,278.37 ± 859.84 μm^2^ in *D. chrysippus*, 11,537.01 ± 699.08 μm^2^ in *D. genutia*, and 14,426.65 ± 647.59 μm^2^ in *D. plexippus.* The cover scales are coronal and mostly with a rounded apex in *D. chrysippus.* They are foliate in *D. plexippus* and *D. genutia*, with a rounded apex in the former and a bifurcated apex in the latter. The size of the cover scales is 11,248.81 ± 521.82 μm^2^ in *D. chrysippus*, 8648.95 ± 337.19 μm^2^ in *D. genutia*, and 10,271.38 ± 647.59 μm^2^ in *D. plexippus.* Both ground and cover scales are significantly larger than the androconial scales in the three *Danaus* species. The density of ventral scales is highest in *D. chrysippus*, followed by *D. genutia*, and lowest in *D. plexippus*, with significant differences (*p* < 0.0001). Additionally, the density of the scent patch scales on the ventral wing surface is significantly lower than that of the androconial scales in each species (Table 2).

The upper surface of the scent patch scales typically consists of a series of lumens framed by right-angled ridges and crossribs at the nano-scale (Figure 5D–L). In the sectional view, the ridges overlap with tree-shaped lamellae, while the crossribs are vertically connected to the lower surface by interlaced trabeculae (Figure 5G–L). The trabeculae of the androconial scales are thickened with small distances, causing the lumens framed by ridges and crossribs to appear covered when viewed from above (Figure 5D–F). In contrast, the trabeculae of the scent patch scales on the ventral wing surface are thin with large distances, resulting in clear lumens when viewed from above (Figure 5J–L). Additionally, compared to androconial scales, the ridges and crossribs in the scent patch scales on the ventral wing surface are narrower and have smaller distances in each species. Height parameters are generally higher in the androconial scales than in the scent patch scales on the ventral wing surface for each species (Table 2).

The androconial scales exhibit significant differences between species in parameters that positively correlate with emissivity, with the highest values mostly found in *D. plexippus*. On the other hand, parameters that negatively correlate with emissivity are the largest in *D. chrysippus* and *D. genutia*. The ridge distance ranges from 1.53 ± 0.03 to 1.92 ± 0.16 μm, with the largest value found in *D. genutia* and the smallest in *D. chrysippus*. The distance between crossribs ranges from 477.50 ± 47.00 to 590.39 ± 54.24 mm, with the largest distance found in *D. chrysippus* and the smallest in *D. genutia*. The widths of the ridges and crossribs range from 327.26 ± 26.97 to 445.61 ± 29.79 nm and 235.45 ± 12.49 to 248.80 ± 20.61 nm, respectively. The ridge width differs significantly among species, while the crossrib width remains constant. All height characteristics are significantly highest in *D. plexippus*, and most of them do not differ significantly in the other two species. The heights of the lower lamina, crossribs, and ridges in different species are all approximately 300 nm with a range of at most around 50 nm for the three species. However, the heights of the lamellae and trabeculae are at least 800 nm with a range of mostly about 200 nm. Nonetheless, the width of the trabeculae in *D. plexippus* is significantly greater than in *D. genutia* (Table 2).

### 3.4. Correlations between the Morphological Characteristics of Antenna and Androconial Scales

When sampling the three species, we found the morphological parameters of the androconial scales to be generally significantly correlated with each other (Figure 6A). The size of the scent patch, which corresponds to the region that the androconial scales cover, increases significantly with the increase in androconial scale size, ridge width, crossrib width, crossrib distance, and trabeculae distance. However, the former two are positively correlated with pheromone emission efficiency, while the latter two are negatively correlated. Additionally, the size of the scent patch increases with the decrease in the height of lamellae, trabeculae, lower lamina, and the width of trabeculae. These parameters are all positively correlated with pheromone emission efficiency. The size of the androconial scales increases significantly as parameters positively correlated with emissivity decrease. Parameters that affect emissivity in the same direction are not always positively correlated with each other. For instance, the width of the crossrib is generally negatively correlated with the height characteristics, although they are all positively correlated with the emissivity. However, most of the height parameters are significantly and positively correlated with each other. It is worth mentioning that these correlations are not consistent across the three species (Figure 6B–D). In some cases, correlations within a species are opposite to those inferred in three species. For instance, in each of the three species, the lower lamina height of the androconial scales and the size of the scent patch are positively correlated with each other. Additionally, some items that are not significantly correlated when the sampling covers all three species are significantly correlated with each other when the sampling is within a species. For example, the width of the ridge is significantly correlated with the area of the scale when the sampling is within one of the species. However, it shows no significant correlation with the scale area when the sampling covers all three species.

The morphometric parameters of the antenna are mainly significantly and positively correlated with each other when the sampling covers the three species (Figure 7A). For example, the length of the antenna increases significantly as the lengths of the scape, pedicel, flagellum, and the area of the apical club increase. The numbers of antennal sensilla are not correlated with each other except for the numbers of ST and Sco, which are positively correlated. The lengths and diameters of the antennal sensilla are correlated with the lengths of antennal segments. Specifically, as the lengths of antennal segments decrease, the lengths of the ST and SCh, as well as the diameter of the Sco, also decrease significantly. Conversely, the length of the Sco and the diameter of the ST increase significantly as the lengths of antennal segments decrease. The lengths and diameters of the antennal sensilla are mostly significantly correlated with each other, except for the diameter of the SCh. These correlations differ between males and females in the three species (Figure 7B,C). For example, in the males, there is a significant negative correlation between the length of Sch and the length of the antenna. However, in the females, there is no significant correlation between these two items. Similarly, there are differences in these correlations across species (Figure 7D–F). For example, in *D. chrysippus*, the area of the apical club shows a positive correlation with the antennal length, whereas in *D. genutia*, the two are negatively correlated. In *D. plexippus*, however, they do not show a significant correlation.

The morphological characteristics of male and female antennae are highly correlated (Figure 8). When the sample includes the three species, an increase in the lengths of antennal segments in one sex generally results in an increase in the number, length, and diameter of antennal sensilla in the other sex (Figure 8A). Likewise, an increase in the length of the ST in males leads to an increase in that in females. These correlations may differ when sampling within a single species. For instance, in *D. chrysippus* and *D. genutia*, there is a positive correlation between males’ and females’ antennal length. However, in *D. plexippus*, this correlation is negative (Figure 8B–D). When the sampling is limited to a single species, there are more correlations between male and female antennal characteristics. For example, the number of SCh per flagellomere in the males is only correlated with the antennal length, flagellomere length, and diameter of Sco in the females when the sampling covers three species, but it is significantly correlated with most antennal parameters in the females when the sampling is within one of the three species.

The parameters of the androconial scales are significantly correlated with those of the females’ antenna (Figure 9A). In the three species, as the lengths of antennal segments in the females decrease, the areas of the androconial scales increase; additionally, the height parameters of the androconial scales, which are positively correlated with scale emissivity, decrease. The areas of scent patches and androconial scales of the males are correlated with the numbers, lengths, and diameters of the females’ antennal sensilla. Specifically, these two items are positively correlated with the lengths of ST and SCh in females but negatively correlated with the length of Sco and the diameters of ST and SCh. The height parameters of the androconial scales of the males are significantly correlated with the lengths and diameters of the females’ antennal sensilla. They are negatively correlated with the lengths of ST and SCh, as well as the diameter of Sco, but positively correlated with the length of Sco and the diameters of ST and SCh in the females. The numbers of ST and Sco in the females decrease significantly as the androconial height parameters increase. When sampling within one of the target species (Figure 9B–D), there are fewer correlations between the characteristics of the males’ scent patches and the females’ antenna. In *D. genutia* and *D. plexippus*, most of these characteristics are uncorrelated with each other.

## 4. Discussion

The present study provides detailed morphological information of the antennae and scent patches for three *Danaus* species, as well as their interspecies and intersexual differences and their correlations. This is fundamental information on the pheromone recognition system of *Danaus*, which not only provides insights into the butterflies’ sexual communication but also knowledge for the development of bionic materials and devices, such as radiative cooling materials and equipment for terahertz technology [12,13,14,15].

This study reports three types of sensilla in the apical antennal club for the three *Danaus* species, which is consistent with the previous study of *D. gilippus* [16]. It can be inferred that the type of antennal sensilla in *Danaus* is consistent. The sensilla’s morphological characteristics, including surface decorations that indicate their function, are similar to previous reports for *D. gilippus* and many other Lepidopteran species, except for STs. In *Danaus* and some butterfly species, such as *Ascia monuste* (Linnaeus) of Pieridae, the great southern white butterfly, it is smooth, but in *Talponia batesi* Heinrich of Tortricidae and *Stenachroia elongella* Hampson of Pyralidae, the cob borer, it has thread-like ridges [25,26]. Surface roughness has been speculated to be essential for detecting external volatiles [27]. Therefore, it can be inferred that these non-porous STs of *Danaus* are mechanoreceptors and not involved in olfactory function. On the other hand, the other two types of sensilla (Sco and SCh) are multiporous and similar to those found in Hesperiidae, Pieridae, and Lycaenidae, which have been inferred to be involved in chemical sensing [28,29,30,31]. Two other types of sensilla have been reported in butterflies, including *D. gilippus*: sensilla squamiformia and Böhn bristles. However, they were only reported from the scape, the pedicel, or the basal part of the flagellum [32,33]. These sensilla were recognized as mechanoreceptors and are unlikely to play a primary role in sexual communication [34].

The distribution of antennal sensilla we observed in the three *Danaus* species is similar to some but not all butterflies. The longitudinal ridges on the antenna, which frame the two regions where the sensilla are clustered, have been described in other butterflies, such as *Vanessa atalanta* (Linnaeus), the red admiral; *Inachis io* (Linnaeus), the peacock butterfly; and *Nymphalis antiopa* (Linnaeus), the mourning cloak [34]. However, the ridge is absent in many species, such as *Parnara guttata* (Bremer & Grey) of the Hesperiidae, the common straight swift [32], suggesting that it is not universal in butterfly antennae. Similarly, the sulci, where the ST clusters occur, do not always appear in pairs. In *A. monuste* (Linnaeus), each flagellomere has only one sulcus on its venter [21]. In other butterfly genera, such as *Parnara* and *Pelopidas* of the Hesperiidae, STs also cluster into square sulci or do not form a sulcus [32]. SCh and Sco are also scattered in the microtrichia of other butterfly species. However, their amounts differ from those in *Danaus* species [28,32]. Most of the antennal parameters also show interspecific differences among the three *Danaus* species, too. The sexual dimorphism of *Danaus* antennae has not been studied. Our observations showed that the three *Danaus* species generally do not show sexual dimorphism in the number and size of sensilla, but the size parameters of antennal segments differ between the sexes.

Quantitative information on the ultrastructure of the *Danaus* scent patch has not been previously reported. However, the morphological characteristics we observed are consistent with relevant studies. The coronal shape and high density of androconial scales have been reported in *D. chrysippus* [8]. The highly thickened cuticle in the scent patches has been found in several butterfly families including Nymphalidae, Hesperiidae, Pieridae, and Lycaenidae [12,13,35,36]. The observed hierarchical structures of the androconial scales, including the coronal shape, the framework at the nano-scale, and the overlapping extent, are consistent with the wing scales in scent patches of other Nymphalidae species [37,38]. The androconial scales are all distinctly smaller than the other scales in all previous reports, too [13,36]. In addition, most scent patch substructures in previous reports show significant differences between species, and these differences do not always have the same effect on pheromone emission efficiency [24,39,40].

There has been no test conducted on the correlation between the morphological characteristics of the antenna and the scent patch, or androconial scales, of butterflies. Here, for the first time, we prove that the morphological characteristics of the antenna and the scent patch are correlated. The size of the scent patch is mainly negatively correlated with factors that can increase pheromone dissipation but positively correlated with factors that can increase the reception of chemical substances in the antenna [24]. This correlation may support the adaptation between the different organs of the copulatory system [8]. The correlation between male and female antenna characteristics suggests intraspecific structural fitness. Additionally, the correlations between the parameters were not consistent across different species, suggesting that the sexual communication system has morphological differentiation at the species level. Therefore, it is recommended to examine the correlations between morphological characteristics at different taxonomic levels when resolving their functional mechanism.

To our knowledge, this is the first quantitative comparative morphological study of the antenna and scent patch of *Danaus* and the first attempt to investigate their correlation. However, knowledge of the morphological characteristics of these two organs in the genus, as well as in other butterflies, is still quite limited. Further studies involving a wider range of taxa are needed to address the functional morphology of the sexual communication system of butterflies.

## Figures and Tables

**Figure 1 insects-15-00121-f001:**
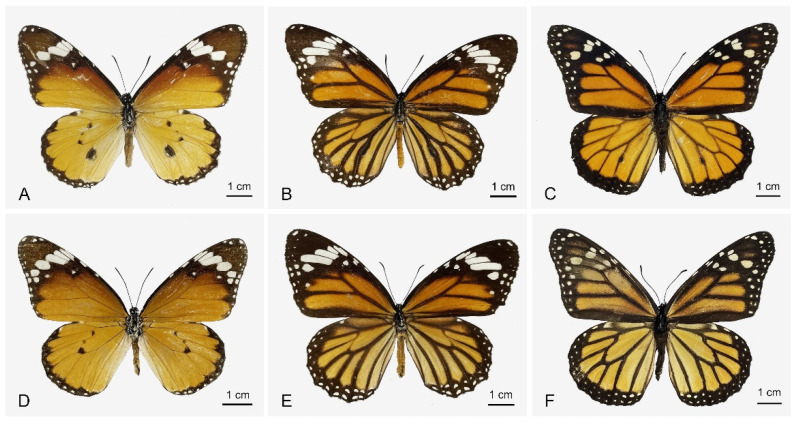
Dorsal view of both sexes of the three *Danaus* species. (**A**,**D**) *D. chrysippus*. (**B**,**E**) *D. genutia*. (**C**,**F**) *D. plexippus*. (**A**–**C**) Males. (**D**–**F**) Females.

**Figure 2 insects-15-00121-f002:**
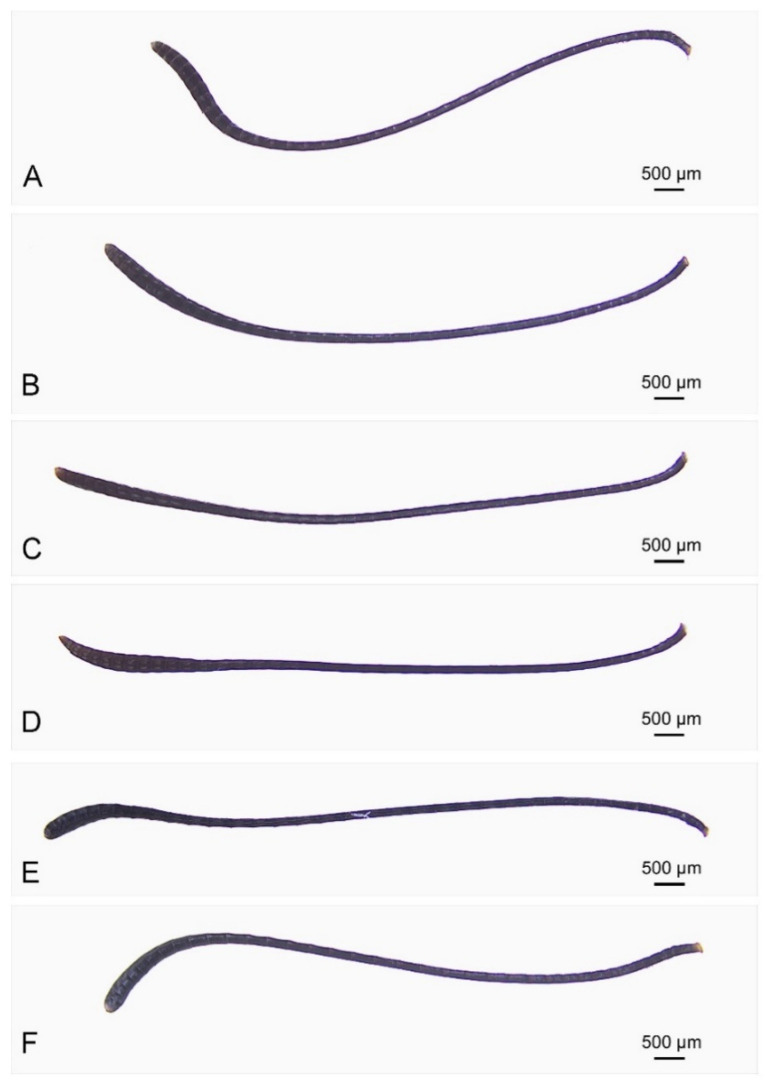
Antennae of both sexes of three *Danaus* species. (**A**,**B**) *D. chrysippus*. (**C**,**D**) *D. genutia*. (**E**,**F**) *D. plexippus* (male and female, respectively).

**Figure 3 insects-15-00121-f003:**
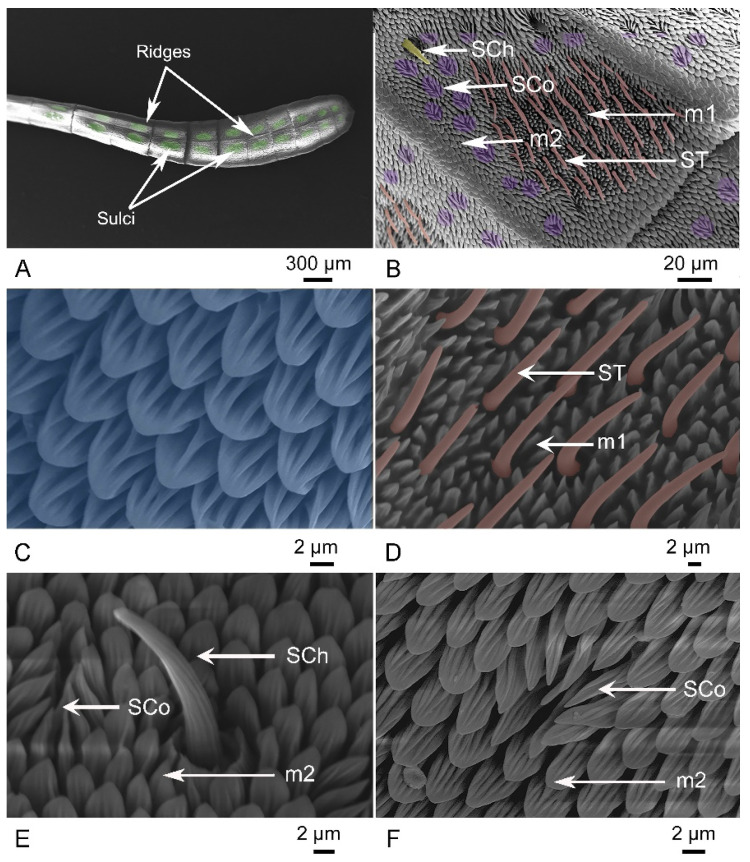
Micrograph of different antennal sensilla of the three *Danaus* species. (**A**) Apical club of flagellum with sulci marked in green. (**B**) Distribution of sensilla trichodea (ST, orange), sensilla chaetica (SCh, yellow), and sensilla coeloconica (Sco, purple), and two subtypes of microtrichia (m1, m2). (**C**) Micrograph of m2. (**D**) Micrograph of ST (orange) and m1. (**E**) Micrograph of SCh, Sco and m2. (**F**) Micrograph of Sco and m2.

**Figure 4 insects-15-00121-f004:**
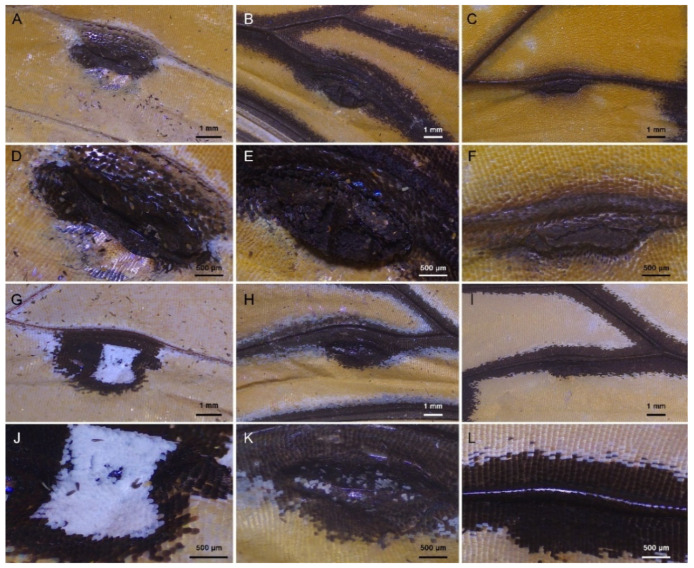
The scent patches of the three *Danaus* species. (**A**–**F**) The scent patches on the dorsal wing surface at different magnifications. (**G**–**L**) The scent patches on the ventral wing surface at different magnifications. (**A**,**D**,**G**,**J**) *D. chrysippus*; (**B**,**E**,**H**,**K**) *D. genutia*; (**C**,**F**,**I**,**L**) *D. plexippus*.

**Figure 5 insects-15-00121-f005:**
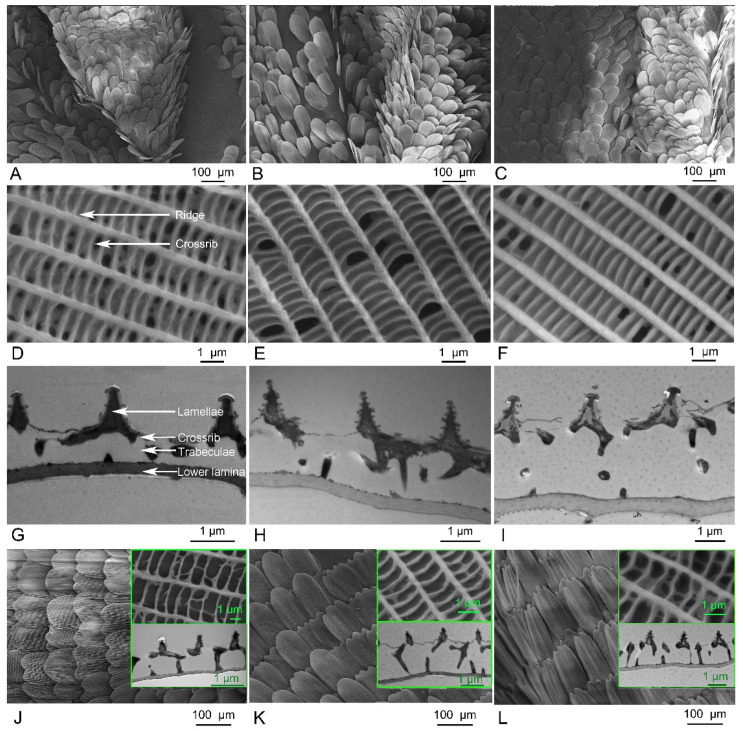
The ultrastructure of scent patch scales in the three *Danaus* species. (**A**–**C**) The surface view of scent patch scales on the dorsal wing surface obtained from the scanning electron microscope. (**D**–**F**) The surface ultrastructure of androconial scales obtained from the scanning electron microscope. (**G**–**I**) The sectional ultrastructure of androconial scales obtained from the transmission electron microscope. (**J**–**L**) The surface and sectional ultrastructure of scent patch scales on the ventral surface of the wings. (**A**,**D**,**G**,**J**) *D. chrysippus*; (**B**,**E**,**H**,**K**) *D. genutia*; (**C**,**F**,**I**,**L**) *D. plexippus*.

**Figure 6 insects-15-00121-f006:**
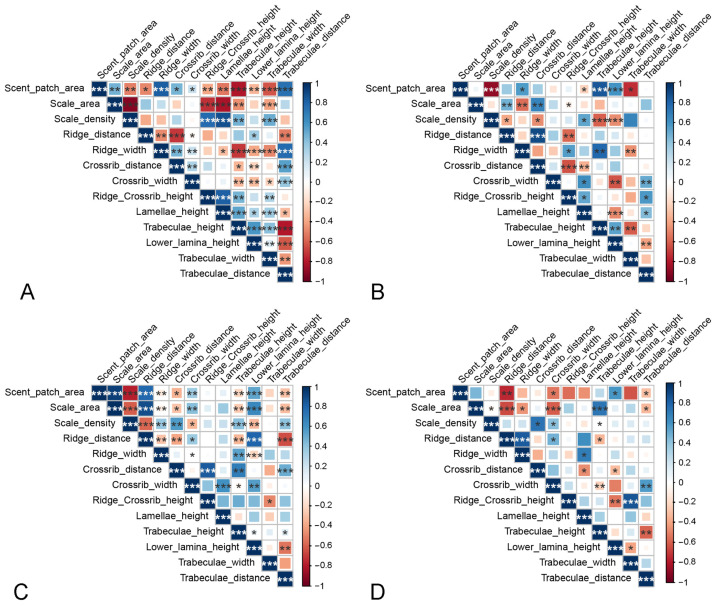
The correlation between the morphological characteristics of androconial scales. (**A**) In the three *Danaus* species. (**B**) In *D. chrysippus*. (**C**) In *D. genutia*. (**D**) In *D. plexippus*. The area and color of the squares indicate the strength of the correlations. The stronger the correlation is, the larger the square and the deeper its color. Squares are in blue if the correlation is positive and in red if the correlation is negative. ‘*’ indicates significance at the 0.05 level. ‘**’ indicates significance at the 0.01 level. ‘***’ indicates significance at the 0.001 level.

**Figure 7 insects-15-00121-f007:**
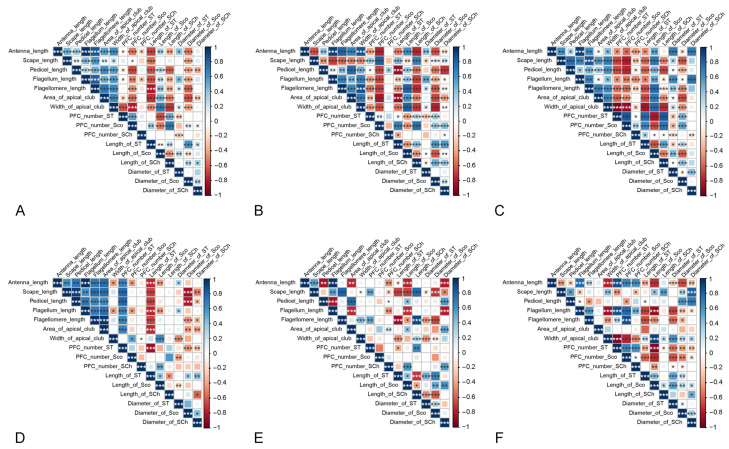
The correlation between morphological characteristics of the antenna. (**A**) In the three *Danaus* species. (**B**) In males. (**C**) In females. (**D**) In *D. chrysippus*. (**E**) In *D. genutia*. (**F**) In *D. plexippus*. The area and color of the squares indicate the strength of the correlations. The stronger the correlation, the larger the square and the deeper its color. Squares are blue when the correlation is positive and red when the correlation is negative. The ‘*’ indicates significance at the 0.05 level. ‘**’ indicates significance at the 0.01 level. ‘***’ indicates significance at the 0.001 level.

**Figure 8 insects-15-00121-f008:**
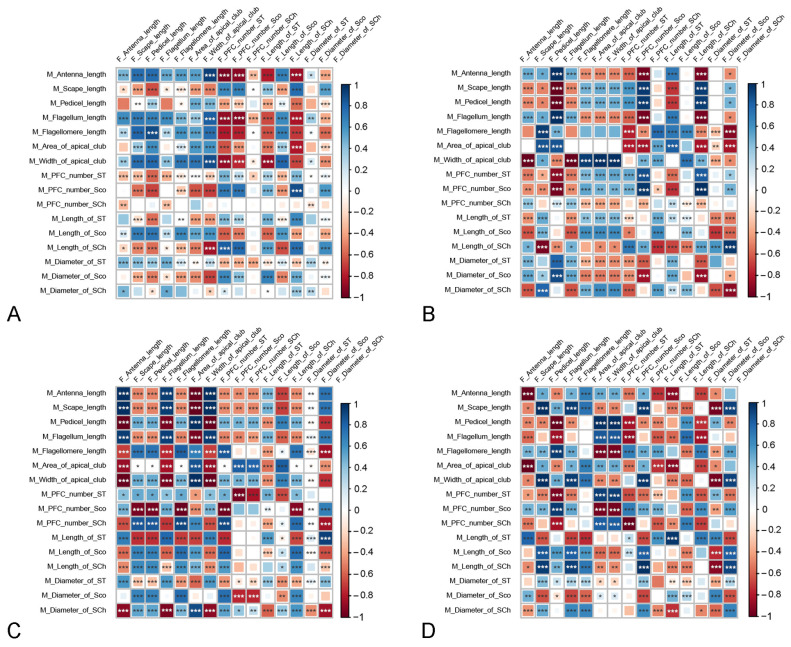
The correlation between the morphological characteristics of male and female antennae. (**A**) In the three *Danaus* species. (**B**) In *D. chrysippus*. (**C**) In *D. genutia*. (**D**) In *D. plexippus*. The area and color of the squares indicate the strength of correlations. The stronger the correlation is, the larger the square and the deeper its color. Squares are in blue if the correlation is positive and in red if the correlation is negative. ‘*’ indicates significance at the 0.05 level. ‘**’ indicates significance at the 0.01 level. ‘***’ indicates significance at the 0.001 level.

**Figure 9 insects-15-00121-f009:**
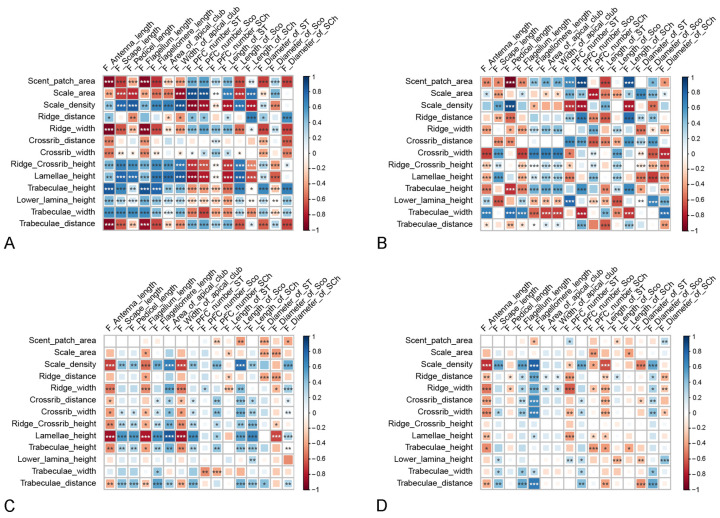
The correlation between the morphological characteristics of the females’ antenna and the males’ androconial scales. (**A**) In the three *Danaus* species. (**B**) In *D. chrysippus*. (**C**) In *D. genutia*. (**D**) In *D. plexippus*. The area and color of the squares indicate the strength of correlations. The stronger the correlation is, the larger the square and the deeper its color. Squares are in blue if the correlation is positive and in red if the correlation is negative. ‘*’ indicates significance at the 0.05 level. ‘**’ indicates significance at the 0.01 level. ‘***’ indicates significance at the 0.001 level.

**Table 1 insects-15-00121-t001:** Morphological features of the antenna of three *Danaus* species and the morphometric measurements of their sensilla types (Mean ± SE).

Item	*D. chrysippus*		*D. genutia*		*D. plexippus*	
	Male	Female	Male	Female	Male	Female
Antenna length (mm)	28.82 ± 0.16 ^Ba^	25.66 ± 0.18 ^Bb^	28.33 ± 0.16 ^Ba^	28.13 ± 0.13 ^Bb^	31.11 ± 0.18 ^Aa^	30.41 ± 0.21 ^Ab^
Scape length (mm)	0.42 ± 0.02 ^Aa^	0.35 ± 0.01 ^Ab^	0.45 ± 0.01 ^Aa^	0.35 ± 0.01 ^Ab^	0.40 ± 0.01 ^Ab^	0.43 ± 0.02 ^Aa^
Pedicel length (mm)	0.41 ± 0.01 ^Aa^	0.32 ± 0.06 ^Ab^	0.29 ± 0.01 ^Ba^	0.30 ± 0.02 ^Ba^	0.33 ± 0.01 ^Ab^	0.37 ± 0.03 ^Aa^
Flagellum length (mm)	27.95 ± 0.24 ^Ba^	24.98 ± 0.18 ^Bb^	27.60 ± 0.16 ^Ba^	27.49 ± 0.15 ^Bb^	30.38 ± 0.18 ^Aa^	29.79 ± 0.04 ^Ab^
Flagellomere length (mm)	0.75 ± 0.01 ^Ba^	0.65 ± 0.02 ^Bb^	0.72 ± 0.01 ^Ba^	0.67 ± 0.03 ^Bb^	0.77 ± 0.02 ^Aa^	0.75 ± 0.01 ^Aa^
Area of apical club (mm^2^)	3.81 ± 0.10 ^Aa^	3.42 ± 0.15 ^Ab^	3.18 ± 0.13 ^Ba^	3.23 ± 0.21 ^Ba^	3.85 ± 0.28 ^Aa^	3.72 ± 0.34 ^Aa^
Width of apical club (mm)	0.85 ± 0.04 ^Ba^	0.86 ± 0.03 ^Ba^	0.75 ± 0.04 ^Ca^	0.73 ± 0.03 ^Cb^	0.98 ± 0.01 ^Ab^	1.02 ± 0.02 ^Aa^
PFC number of ST	121 ± 2 ^ABb^	130 ± 6 ^ABa^	131 ± 6 ^Aa^	129 ± 5 ^Aa^	127 ± 6 ^Ba^	114 ± 1 ^Bb^
PFC number of Sco	285 ± 16 ^Ba^	291 ± 9 ^Ba^	347 ± 19 ^Aa^	336 ± 27 ^Aa^	282 ± 17 ^Ba^	261 ± 4 ^Bb^
PFC number of SCh	7 ± 1 ^Aa^	7 ± 1 ^Aa^	7 ± 1 ^Aa^	7 ± 1 ^Aa^	7 ± 1 ^Aa^	6 ± 1 ^Aa^
Length of ST (μm)	17.40 ± 0.36 ^Bb^	20.99 ± 0.63 ^Ba^	19.36 ± 0.51 ^Ab^	21.87 ± 0.49 ^Aa^	18.04 ± 0.46 ^Bb^	18.83 ± 0.71 ^Ba^
Length of Sco (μm)	16.10 ± 0.47 ^Ba^	16.33 ± 0.72 ^Ba^	15.49 ± 0.30 ^Bb^	16.15 ± 0.48 ^Ba^	16.68 ± 0.68 ^Ab^	18.34 ± 0.28 ^Aa^
Length of SCh (μm)	26.47 ± 1.43 ^Ba^	25.98 ± 1.47 ^Ba^	34.20 ± 1.21 ^Aa^	30.72 ± 1.56 ^Ab^	25.33 ± 1.92 ^B^	25.38 ± 0.87 ^B^
Diameter of ST (μm)	1.79 ± 0.13 ^Ba^	1.77 ± 0.08 ^Ba^	1.82 ± 0.08 ^ABa^	1.90 ± 0.10 ^ABa^	1.89 ± 0.06 ^Ab^	1.96 ± 0.06 ^Aa^
Diameter of Sco (μm)	8.40 ± 0.31 ^ABb^	8.96 ± 0.12 ^ABa^	9.27 ± 0.19 ^Ab^	9.26 ± 0.23 ^Aa^	8.27 ± 0.43 ^Bb^	8.82 ± 0.42 ^Ba^
Diameter of SCh (μm)	3.53 ± 0.17 ^Ba^	3.76 ± 0.22 ^Ba^	4.83 ± 0.18 ^Aa^	4.93 ± 0.08 ^Aa^	4.69 ± 0.10 ^Ab^	4.90 ± 0.20 ^Aa^

Notes. Same capital letters indicate no significant differences between species (*p* > 0.05). Same lower-case letters indicate no significant differences between sexes (*p* > 0.05). PFC: per flagellomere of the apical club.

**Table 2 insects-15-00121-t002:** Morphological features of scent patch scales (including the androconial scales) of the three *Danaus* species (Mean ± SE).

Item	Androconial Scales on the Dorsal Wing Surface			Scent Patch Scales on the Ventral Wing Surface		
	*D. chrysippus*	*D. genutia*	*D. plexippus*	*D. chrysippus*	*D. genutia*	*D. plexippus*
Density (number/mm^2^)	557.91 ± 11.99 ^Ba^	400.15 ± 9.19 ^Ca^	643.32 ± 4.78 ^Aa^	348.72 ± 14.04 ^Ab^	320.51 ± 8.88 ^Bb^	263.25 ± 7.50 ^Cb^
Ridge distance (μm)	1.53 ± 0.03 ^Cb^	1.92 ± 0.16 ^Aa^	1.71 ± 0.09 ^Bb^	1.81 ± 0.14 ^Ba^	1.94 ± 0.13 ^ABa^	2.08 ± 0.18 ^Aa^
Ridge width (nm)	445.61 ± 29.79 ^Aa^	363.92 ± 19.38 ^Ba^	327.26 ± 26.97 ^Ca^	281.34 ± 33.77 ^Bb^	332.76 ± 27.07 ^Ab^	256.75 ± 15.45 ^Bb^
Crossrib distance (nm)	590.39 ± 54.24 ^Aa^	477.50 ± 47.00 ^Bb^	519.55 ± 30.54 ^Bb^	616.65 ± 78.79 ^Ba^	872.06 ± 75.04 ^Aa^	709.15 ± 72.33 ^Ba^
Crossrib width (nm)	248.80 ± 20.61 ^Aa^	235.69 ± 14.97 ^Aa^	235.45 ± 12.49 ^Aa^	164.13 ± 12.56 ^Bb^	220.01 ± 14.77 ^Ab^	125.08 ± 15.35 ^Cb^
Lamellae height (μm)	0.84 ± 0.07 ^Bb^	0.73 ± 0.06 ^Ca^	1.74 ± 0.21 ^Aa^	0.99 ± 0.14 ^Aa^	0.62 ± 0.19 ^Ba^	0.64 ± 0.07 ^Bb^
Ridge/crossrib height (nm)	339.69 ± 51.53 ^Ba^	264.33 ± 41.77 ^Cb^	405.97 ± 42.57 ^Aa^	292.55 ± 27.00 ^Ab^	313.47 ± 30.05 ^Aa^	247.17 ± 35.46 ^Bb^
Trabeculae height (μm)	1.02 ± 0.17 ^Cb^	1.41 ± 0.22 ^Bb^	2.17 ± 0.28 ^Aa^	2.29 ± 0.18 ^Aa^	1.71 ± 0.21 ^Ba^	1.64 ± 0.27 ^Bb^
Lower lamina height (nm)	312.17 ± 38.96 ^Ba^	326.43 ± 29.62 ^Ba^	362.87 ± 35.13 ^Aa^	137.39 ± 9.76 ^Bb^	187.67 ± 21.25 ^Ab^	190.82 ± 31.83 ^Ab^
Trabeculae width (nm)	229.65 ± 33.81 ^Ba^	237.02 ± 23.24 ^Bb^	273.57 ± 24.47 ^Aa^	137.60 ± 18.83 ^Bb^	272.53 ± 44.51 ^Aa^	129.37 ± 21.73 ^Bb^
Trabeculae distance (nm)	969.31 ± 117.84 ^Ab^	784.08 ± 76.61 ^Bb^	652.74 ± 117.15 ^Ab^	1785.40 ± 280.41 ^Aa^	1209.47 ± 131.21 ^Ba^	798.00 ± 81.29 ^Ca^

Notes: same capital letters indicate no significant differences in the target scale type between species (*p* > 0.05). Same lower-case letters indicate no significant differences between the two types of scales (*p* > 0.05).

## Data Availability

The data presented in this study are openly available in FigShare at http://doi.org/10.6084/m9.figshare.24500983 (accessed on 3 February 2024).

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
