# Peer review of "Morphological Characterization of the Antenna and Scent Patch of Three Danaus Species (Papilionoidea: Nymphalidae, Danainae)"

_insects, 2024, doi:10.3390/insects15020121_

Round 1
Reviewer 1 Report (Previous Reviewer 1)
Comments and Suggestions for Authors
Abstract need to improve in some parts. Introduction section is improved significantly but still some modification needed (add citation for specific information and some missed details). Few corrections were recorded in materials and methods section. Results section should be improved especially at the first paragraph. Also initial paragraph should explain comparative morphology for both sexes of each species must be added as first paragraph in results section (Fig.1…this not mentioned at all in the text). Title of tables should be changed. In figures 3 a labelling for sensillum inside the micrographs should be added especially Fig.3 D that explore two different types of sensillum. Corrections for sensillum description was also made (see sticky notes). Some sentences are make confusion for readers that need to reconstruct. Lines 241-242 and 262-264, I think there are a mistakes in orientation of these lines that should be deleted. Discussion section need improvement
Detailed comments were included in the attached PDF file with tracking mode that could be shared with the authors.

Authors have done a good job but still some sentences are unclear or confused because of errors in English style (too long sentences or need reconstruct) and grammar errors.
Author Response
Please see the attachment.

Reviewer 2 Report (Previous Reviewer 2)
Comments and Suggestions for Authors
This is a much better version. However I am not satisfied with the chapter 3.2, what needs a profound revision.
The characterization of Danaine scent patches are confusing. There are no dorsal and ventral surfaces, but dorsal and ventral wing surface (alar) organs (lines 241-255). It seems that not the scent patches were characterized but a single type of scale, what was supposedly a kind of androconium, and the dimensions of that were correlated with flagella dimensions (see the next entry).
Table 2 is a not the one what is indicated: there is no scent patch characterization, but a kind of scale is dimensioned (lines 260-268). There is several kinds of scales in and around the scent patches. These should be carefully explored, characterized and the scales which disseminate the aphrodite scent that should be selected for the comparative approach.
In Figure 5 it is not clear from where the scales are taken from (dorsal or the ventral organ? from which part?) and there is no evidence provided that they figured scales are androconia indeed. The quality of the Figs 5A-C is poor, they should be replaced; Figs 5D.E are too dark, etc.
Figure 6 is very difficult to read.
In the title write "Papilionoidea" instead of "Papilioidea).
Comments on the Quality of English LanguageSome English smoothing would be still necessary.
Round 2
Reviewer 1 Report (Previous Reviewer 1)
Comments and Suggestions for Authors
I appreciate how the authors were able to revise the manuscript accordingly, thereby improving its quality and readability very significantly. The authors have successfully addressed my initial comments. Some minor revisions are hereby suggested, in addition to some points that need clarification/reconsideration

Some corrections are suggested throughout the text, in addition to some sentences that need clarification/reconsideration (highlighted in PDF)
Author Response
Thank you for the valuable comments, please see the attachment.

Reviewer 2 Report (Previous Reviewer 2)
Comments and Suggestions for Authors
This is a much better and improved version. It gives a lot of technical details therefore its interest is limited. It is a kind of well documented data set and analysis, what may be suitable for further reserach and investigations in biomimetcs. Some annotations are listed below. Line numbers are used as references.
General. There is little difference between the Simple Summary and the Abstract.
15 - in scent patches (instead of "on scent patches") [there are more similar cases in the manuscript, these should be checked carefully]
27 - roof-tree nanonstrucure?? the scales are arranged on the wing membrane as tiles on the roof (in micron realm), and certain scale type has tree-like (Morpho-type) structure in its lumen (in nano realm)
40 - Danaus Kluk, 1780 (instead of "Danaus Kluk, 1804")
83 - investigate (instead of "investigated")
100 - were identified on the basis of wing colouration and pattern (instead of "were identified according to taxonomic characterization")
105 - were investigated (instead of "were observed")
108 - replicates? = specimens?
159 - The chapter 3.1. does not give morphological results, but describes the wing pattern and colour characteristics of the wings. The subtitle should be changed accordingly.
175 - in their general appearance (instead of "in their general morphology")
163 - but black in D. chrysippus (instead of "but black in D. genutia")
Comments on the Quality of English Language
There are some little linguistic problems, for example mixing prepositions "in" and "on", misusing the term "morphology", etc. a scientific editor should work on the manuscript
Author Response
Thank you for the valuable comments, please see the attachment.

This manuscript is a resubmission of an earlier submission. The following is a list of the peer review reports and author responses from that submission.
Round 1
Reviewer 1 Report
Comments and Suggestions for Authors
The manuscript does present information on a number of morphological and morphometric characteristics of the antenna and the scent patch of three Danaus species. Anatomical or anatomy word must be deleted from all text, it is ultrastructure characterization. I would consider changing the title. The manuscript has extensive grammatical and syntactical issues that need to be addressed. I would strongly recommend English editing or revision preferably native speaker, who could correct grammatical mistakes, typos and improve the style of the manuscript. I corrected some grammar errors, and I also found quite a lot of stylistic errors, such as tautology and unnecessary (in my opinion) repetitions. I also think that the abstract should be corrected significantly. Introduction must be improved. Materials and Methods section need to improve via adding some missed details. Results need improvement and provided new SEM images with good quality (the current photos is not qualified: need adding new photos and replace some with high resolution). I did not find TEM photos informative or add significant value for the study. Critical point in results section is: 1- Fig. 1 must contain general view for whole antennae for male and female for each species: Authors must add photos for whole antennae for male and female for each species. Moreover, bad resolution for Fig .1A and missed labelled for each magnified sensillum type.
2- Identification of sensillum types must be corrected: ex. In Figure 1 (it is Sensilla basiconica not Sensilla trichodea). I encourage authors to read updated published paper on identification and description of sensillum types (recommend some in PDF file).
A. The wide range between the difference in length in table 1 indicate it should be different sensillum types or subtypes
B. The wide range between the difference in length indicate it should be different sensillum types or subtypes
C. Missed information and photos for scape and pedicle sensillum types.
3-Tables title must be corrected.
Discussion section is rather weak and must be improved.
Detailed comments were included in the attached PDF file with tracking mode that could be shared with the authors.

The manuscript has extensive grammatical and syntactical issues that need to be addressed. I would strongly recommend English editing or revision preferably native speaker, who could correct grammatical mistakes, typos and improve the style of the manuscript. I corrected some grammar errors, and I also found quite a lot of stylistic errors, such as tautology and unnecessary (in my opinion) repetitions.
Reviewer 2 Report
Comments and Suggestions for Authors
This would be an interesting paper, with good data on Danaus micromorphology and quantitative analyses, but I do not recommend for publication. My reasons are birefly outlined below.
1. Danaus androconia have been intesively studied already by classical workers, including the subjects of this paper D. chrysippus, D. genutia and D. plexippus. They are not at all poorly known. In the introduction this fact is misinterpreted.
2. Any genus, including Danaus in general cannot have population decline, but certain Danaus species may have indeed, and on the basis on that the species can be declared as "endangered". Therefore it is a constrained goal to prove the necessity of studing anatomical characterstics on such statement. Moreover the courthsip of the different species are different in many details and because of the uncountable variuables, it is rather difficult to apply any "androconia" results for conservation purpose. As the discussion suggests, the goal of the investigation would be more straightforward to find directions and corraletions for biomimetic applications. This should be addressed already in the introduction.
3. The authors communicate that they examined three Danaus species, namely D. chrysippus, D. genutia and D. plexippus (the imagines should be shown for positive identifications). However the last mentioned species does not occur not only in the last twenty years, but never has been established natural colonies in China. The D. plexippus material the authors examined came from South America, Amazonas, Tingo Maria (most probably from commercial stock). The D. plexippus populations of the Amazonas basin is non-migratory and represent the taxon D. p. megalippe (Hübner). The fact that the species never occured in China, and the identity of D. plexippus sensu authors is not clear, is enough to reject the paper in its present form.
4. I recommend (a) to give an overview of Danainae research in the intro; (b) the conservation aspect should be deleted and the intro should be sharpened for having biomimetical applications; (c) instead of Danaus plexippus, another Danaus species common and native in subtropical Asia (D. affinis, D. melanippus) should be examined, (d) the quantitive aspect of the methods must be underlined and probably that is the only novel aspect of the manuscript regarding Danainae androconia studies; (e) having better electron scanning micrographs (eg. Fig. B-C show overcharged samples).
Comments on the Quality of English Language
- some places there is no agreement of tenses (cf. lines 41-43)
- style is poor (cf. lines 86-88: the word "collected" occurs four times)
